# Floe-size distributions in laboratory ice broken by waves

Agnieszka Herman[1], Karl-Ulrich Evers[2], and Nils Reimer[2]

[1]Institute of Oceanography, University of Gdansk, Poland
[2]Arctic Technology, Hamburgische Schiffbau-Versuchsanstalt GmbH (Hamburg Ship Model Basin), Hamburg, Germany

*Correspondence to:* A. Herman (oceagah@ug.edu.pl)

**Abstract.** This paper presents the analysis of floe-size distribution (FSD) data obtained in laboratory experiments of ice breaking by waves. The experiments, performed at the Large Ice Model Basin (LIMB) of the Hamburg Ship Model Basin (Hamburgische Schiffbau-Versuchsanstalt, HSVA), consisted of a number of tests in which an initially continuous, uniform ice sheet was broken by regular waves with prescribed characteristics. The floes' characteristics (surface area; minor and major axis, and orientation of equivalent ellipse) were obtained from digital images of the ice sheets after five tests. The analysis shows that although the floe sizes cover a wide range of values (up to 5 orders of magnitude in the case of floe surface area), their probability density functions (pdfs) do not have heavy tails, but exhibit a clear cut-off at large floe sizes. Moreover, the pdfs have a maximum that can be attributed to wave-induced flexural strain, producing preferred floe sizes. It is demonstrated that the observed FSD data can be described by theoretical pdfs expressed as a weighted sum of two components, a tapered power law and a Gaussian, reflecting multiple fracture mechanisms contributing to the FSD as it evolves in time. The results are discussed in the context of theoretical and numerical research on fragmentation of sea ice and other brittle materials.

## 1 Introduction

Recent years have witnessed increasing interest of the sea ice research community in topics related to the floe-size distribution (FSD). A number of new studies are devoted to observational FSD data obtained from airborne and satellite imagery of sea ice (e.g., Perovich and Jones, 2014; Gherardi and Lagomarsino, 2015; Geise et al., 2016; Toyota et al., 2016; Wang et al., 2016), enhancing earlier observations (Inoue et al., 2004; Toyota et al., 2006, 2011; Lu et al., 2008; Steer et al., 2008, among others). Statistical fracture models have been proposed attempting to explain the properties of probability density functions (pdfs) obtained from that data (e.g., Herman, 2010; Toyota et al., 2011; Gherardi and Lagomarsino, 2015). Substantial effort has been made to develop parameterizations of FSD-related processes for numerical sea ice models (Dumont et al., 2011; Williams et al., 2013, 2017; Bennetts et al., 2017). Equations for the evolution of FSD in time, suitable for continuum sea ice models, have been developed by Zhang et al. (2015); Horvat and Tziperman (2015) derived more general equations for joint floe-size and -thickness distribution (see also Horvat and Tziperman, 2017). This increasing interest results from growing evidence that the FSD is a signature of dynamic and thermodynamic processes acting on the ice cover (e.g., Perovich and Jones, 2014; Gherardi and Lagomarsino, 2015; Horvat and Tziperman, 2017), and, presumably even more importantly, that these processes themselves are significantly affected by the floe-size distribution. In short, mutual interactions between FSD

and physics and dynamics of the upper ocean, lower atmosphere and sea ice itself have to be taken into account in order to understand and predict short-term, synoptic and long-term evolution of this complex system.

In spite of substantial progress, many controversies regarding the interpretation of the available FSD data – including the shape of these pdfs – remain unsolved due to the lack of understanding of mechanisms that contribute to the formation of FSD under different conditions. In a great majority of studies, scale-invariance of floe sizes is assumed *a priori* and accordingly, different versions of power-law pdfs are fitted to observational data (tapered or truncated power laws, two power-law regimes separated by a sudden change of slope, etc.). Deviations from power laws are often explained with finite-size effects, i.e., limited spatial resolution and/or extent of images used to determine the FSD, but they can also be produced by physical processes affecting the FSD, e.g., lateral melting/freezing (Perovich and Jones, 2014). In many cases, no convincing arguments for assuming power-law FSDs exist, except the fact that the floe sizes cover a wide range of values. Typically, no alternative pdfs are considered, no measures of the fit errors are provided, and no methods different than least-square fitting of a straight line to a log–log plot of a cumulative floe-size distribution (cdf) is considered – in spite of the fact that this method has a number of well known shortcomings (see, e.g., Clauset et al., 2009; Virkar and Clauset, 2014, for a discussion of typical problems with this approach, including the tendency to produce large systematic errors in the estimated exponents, strong influence of binning on the results, and difficulties with obtaining reliable error estimates).

An example of the process leading to narrow FSDs, with preferred floe sizes, is ice breaking by waves, which is one of the dominating ice fragmentation mechanisms in the marginal ice zone (MIZ). It is still disputable whether the size of ice floes formed in this process depends on wavelength (as assumed by many parameterizations, see Williams et al., 2013, 2017) or rather on material properties and thickness of the ice (as proposed by Squire et al., 1995) – but wave-induced fracturing unquestionably imposes an upper limit on the floe sizes: floes larger than this limit are broken by tensile stresses related to flexural strain. In their recent numerical model of ice breaking by waves, Montiel and Squire (2017) obtained narrow, unimodal pdfs of floe sizes that they describe as "nearly normal". Similar FSDs were obtained with the coupled discrete-element–hydrodynamic model of Herman (2017) when it was run with random variations of ice thickness or strength (unpublished results). In combination with other breaking mechanisms, melting etc., FSDs observed in MIZ may still be (and often are) very wide, but one cannot expect to find scale-invariance in the range of large floe sizes. Accordingly, attempts to fit a power law to the tail of the FSD from MIZ are unjustified, even if a straight line seems to provide a nice fit to a graphic representation of that FSD. The data presented in this paper provide a good illustration of this fact. The results show also the (quite obvious, but often disregarded) fact that limiting the FSD analysis to log–log plots of the respective cdfs provides a distorted and misleading picture of the properties of the respective FSD.

In this work, we present the results of two groups of ice breakup tests performed in 2015 and 2016 in the Large Ice Model Basin (LIMB) of the Hamburg Ship Model Basin (Hamburgische Schiffbau-Versuchsanstalt, HSVA). The tests belong to the first experiments specifically devoted to observing ice breaking by waves under controlled, laboratory conditions. The data collected are used to analyze the FSDs resulting from breaking of initially continuous ice sheets by regular waves with prescribed characteristics. We present floe-size data obtained from digital images of the broken ice sheets, from five test runs. The pdfs of floe sizes are wide (up to 5 orders of magnitude of floe surface area) and have nontrivial shapes, excellently

illustrating typical problems with interpretation of FSD data. We show that the method of presentation of the data – in terms of pdfs of binned data, cdfs of unbinned data, and so on – may influence data interpretation by accentuating certain aspects and obliterate others. We fit the observed pdfs with a function that is a weighted sum of two probability distributions, a tapered power law and a Gaussian, we discuss theoretical arguments underlying this choice of pdf, and interpret the obtained values of the fitted parameters.

The paper is structured as follows: section 2 provides a description of the research facility and of the two groups of experiments (section 2.1), as well as image processing methods used and the collected floe-size data (section 2.2). In section 3, after a short analysis of floe shapes and orientation, a theoretical probability distribution function that combines a tapered power law with a normal distribution is proposed and fitted to the experimental data. Section 4 provides a discussion of the results in view of theoretical research on fragmentation of brittle materials and finishes with conclusions.

## 2   Experiment setup and data

As already mentioned, the measurements described in this paper have been collected during two groups of tests performed at HSVA within two different projects. The first tests – denoted Test Group A further on – were performed by the HSVA researchers as a proof of concept, i.e., they were carried out in a very simple setting, with only few most crucial instruments installed. The second set of experiments (Test Group B) was part of the Hydralab+ Transnational Access project "Loads on Structure and Waves in Ice" (LS-WICE; Hydralab+ project under the Horizon 2020 EU-Framework programme for research and innovation, H2020-INFAIA-2014-2015), performed by an international group of scientists from Norway, USA, Poland and Germany (see Cheng et al., 2017; Herman et al., 2017; Tsarau et al., 2017, for preliminary results). In LS-WICE, a large set of instruments was used, measuring the wave characteristics at several locations along the ice tank, as well the motion of the ice itself. In both test groups, the progress of breaking was continuously recorded on video, and digital images of the ice sheet were taken at selected time instances, as described further in this section.

Crucially for the interpretation of the results, only one ice sheet per each test group was used, i.e., in both cases the experiment started with a continuous ice sheet, and the successive tests were initialized with ice broken in the previous ones. In other words, in each of the two test groups, only the first breaking event took place in a previously intact ice sheet.

### 2.1   Description of the facility and experiments

#### 2.1.1   The Large Ice Model Basin

The ice tank at LIMB is 72 m long, 10 m wide and 2.5 m deep over most of its length, with a deep water (5 m) section for $x \geq 60$ m. (In the remaining part of the paper, all positions are given in a Cartesian coordinate system with origin at the lower left corner of the tank when viewed as in Fig. 1, with the $x$ axis directed along the tank and the $y$ axis directed across the tank.) In both groups of experiments, the waves were generated with four flap type mobile wave generator modules that covered the total 10 m width of the ice tank and were located at $x = 2$ m (Evers, 2017). In Test Group B, a parabolic-shaped beach

was mounted at $x = 70$ m, designed specifically for this project in order to minimize wave reflection (Cheng et al., 2017). No similar device was mounted in the tank in Test Group A, but due to shorter waves and stronger attenuation in those tests (see further), the amount of wave energy reaching the downwave end of the tank was insignificant.

According to the standard procedure at HSVA, ice sheets were produced by seeding under air temperature of approximately $-22°$C (Evers, 2017). The water salinity equaled 6.8 PSU, and the salinity of the ice 3.5–3.8 PSU. The average rate of ice thickness growth was 2 mm/hour. During the experiments, the air temperature was increased towards $0°$C to avoid undesired freezing of open-water areas, ice formation on instruments etc. In normal operating conditions at LIMB, ice formation in front of the wavemaker (trim tank area) is prevented by an insulating sliding gate located at $x = 11.5$ m. During LS-WICE (Test Group B), due to failure of this wall, ice formed over the entire surface area of the tank and was manually removed from the trim tank region before the tests, and the ice edge was located at $x = 20$ m. In both test groups, narrow strips of ice ($\sim$10 cm) were removed from both sides of the ice sheet to reduce the influence of the side walls on wave propagation and ice breaking.

The facility is equipped with a downward-looking camera mounted on a crane that can move over the entire tank. Photographs of the ice sheet taken with this camera several times during the Test Groups A and B were used in this work to obtain the floe-size distributions. Table 1 provides a summary of all test runs, with wave parameters used and short information on ice behavior. Note that the wavelengths in Table 1 are open-water wavelengths. The analysis of the sensor data from the LS-WICE experiments shows that the wavelengths within the ice were in the range (0.95,1.05) of those in open water, depending on floe size (Hayley Shen, personal communication). Measurements of the ice properties in each test group were taken after the ambient temperature was increased towards $0°$C, in order to obtain values representative for the conditions during the tests. For details of the procedures used at HSVA to measure ice density, salinity, bending strength, and elastic modulus, see Evers (2017).

### 2.1.2 Test group A

In these tests, no wave measuring equipment was used except a series of 35 markers of the Qualisys Motion Capture System, placed on the ice along the middle line of the tank ($y = 5$ m) from the ice edge (initially at $x = 11.5$ m) up to $x = 23$ m (Fig. 1a). The markers were removed after initial breaking of the ice (i.e., after test 2020) in order to prevent them from getting wet and drowning. Thus, no wave data were collected afterwards, and the only information recorded (apart from the photographs from the crane camera, mentioned above) were videos showing the progress of breaking. The ice thickness $h_{\mathrm{ice}}$ equaled 30 mm, its elastic modulus $E = 9$ Mpa, and bending strength $\sigma_{\mathrm{crit}} = 47.8$ kPa.

Four out of five tests in this group were conducted with short waves ($L \sim 2.5$ m; Table 1). The ice began to break at wave height $H = 5$ cm (test 2020). Breaking started close to the ice edge and gradually progressed up to $x \sim 34$ m. In spite of increasing wave height, the width of the zone of broken ice remained approximately constant throughout tests 2030 and 2050 (only towards the end of 2050, a few new cracks developed downwave of $x \sim 34$ m). This fact was related to strong attenuation of wave energy. The attenuation rate estimated from Qualisys data equaled $3.7 \cdot 10^{-2} \mathrm{m}^{-1}$ in test 2010 and $3.3 \cdot 10^{-2} \mathrm{m}^{-1}$ in test 2020 (see Supplementary Fig. 1). Assuming that these values did not change significantly downwave from the region where the Qualisys markers were installed, the wave height at $x \sim 34$ m was less than 50% of that at the ice edge. It is reasonable to

assume that after the onset of breaking, i.e., in tests 2030 and 2050, the attenuation was even stronger, especially within the zone close to the ice edge, where the relatively small ice floes were undergoing frequent collisions, intense overwash, and even rafting (for the effects of these processes on wave attenuation, see, e.g., Bennetts and Williams, 2015). After the end of test 2050, many areas of the ice sheet were very "worn out", with a layer of slush filling spaces between floes. In the last test, 2060,

attenuation was weaker due to larger wavelength, so that in spite of the same wave height as in the previous run, breaking took place over the whole tank length. The photograph of the ice after test 2060 is shown in Supplementary Fig. 3.

### 2.1.3  Test group B

The ice thickness $h_{ice}$ in the experiments in Test Group B, measured at a number of locations in the tank, varied between 32.5 and 38.5 mm, with an average of 34.8 mm; the ice elastic modulus $E$ equaled 16 MPa; the bending strength $\sigma_{crit}$ varied from

41.5 kPa close to the ice edge to 67.1 kPa in the area close to the beach. The locations of the pressure and ultrasound sensors used in this group of tests is shown in Fig. 1b, together with the locations of five Qualisys markers that were placed on the ice along the central axis of the tank, ~1.5 m apart from each other. Large parts of the ice sheet were continuously monitored with an AXIS camera mounted at the ceiling and two sideward-looking GoPro cameras mounted at the walls.

Contrary to the expectations, in this test group we did not observe progressive breaking starting from the ice edge. Instead,

during test 1440, the ice sheet first broke approximately in the middle of its length, most likely due to effects related to wave reflection. Due to much longer waves than in Test Group A, attenuation within the ice sheet was much weaker (as data from the pressure sensors clearly show; see Supplementary Fig. 2), and in spite of the beach significant wave reflection was present. As discussed in Herman et al. (2017), the first major crack formed shortly after the reflected wave arrived at its location. Even though it cannot be ruled out that some initial, unnoticed flaws in the ice sheet were responsible for the formation of this crack,

it seems clear that once it formed, it had a profound influence on the subsequent development of fractures during tests 1450, 1500 and 1510. For example, during 1450, breaking was much more intense downwave from this crack than upwave, as if it acted as a secondary ice edge (Herman et al., 2017). Supplementary Fig. 3 shows the photograph of the entire tank after test 1510.

### 2.1.4  Note on scaling and test parameters

Before analyzing the results, it is useful to relate the range of wave and ice parameters used in the laboratory to the corresponding "real-world", unscaled conditions. For the wavelengths $L$ used in the tests, $kh_{ice}$ varied between 0.054 and 0.075 in Test Group A and between 0.035 and 0.097 in Test Group B (Table 1; $k = 2\pi/L$ is the wavenumber). For an unscaled ice thickness of, say, 1.5 m, typical for example for first-year sea ice in the Southern Ocean, those values of $kh_{ice}$ correspond to waves with deep-water lengths of 126–175 m and periods 9.0–10.6 s in Test Group A. In Test Group B the range is wider,

with 97–266 m and 7.9–13.1 s, respectively. In both cases these are realistic wind-wave conditions in the marginal ice zone. The observed attenuation rates $\alpha_a$ (Supplementary Figs. 1 and 2) are within the range of observed ones as well. For example, in test 2010 $\alpha_a = 3.7 \cdot 10^{-2}$ m$^{-1}$, $L = 2.5$ m, and the corresponding attenuation rate for the unscaled wavelength of 126 m is $7.3 \cdot 10^{-4}$ m$^{-1}$. An analogous value from test 2020 is $6.5 \cdot 10^{-4}$ m$^{-1}$, and for the longer waves in tests from Group B:

$9.3 \cdot 10^{-5}$ m$^{-1}$. These values are within the range of those reported in the literature, see, e.g., Kohout and Meylan (2008) or Williams et al. (2013).

Another important aspect of the test setup is related to the mechanical properties of the ice. The values of $E$ and $\sigma_{\mathrm{crit}}$ give a rough estimate of the fracture flexural strain $\epsilon_{\mathrm{crit}}$. For Test Group A $\epsilon_{\mathrm{crit}} = 5.3 \cdot 10^{-3}$, for Test Group B $\epsilon_{\mathrm{crit}}$ varied from $2.6 \cdot 10^{-3}$ at the ice edge to $4.2 \cdot 10^{-3}$ close to the beach. These values are much higher than those typically observed in the field ($5 \cdot 10^{-5}$–$1 \cdot 10^{-4}$), mainly due to very low elastic modulus of the laboratory ice. Consequently, as the maximum strain in sine waves is given by $\epsilon_{\mathrm{max}} = h_{\mathrm{ice}} a k^2 / 2$ (where $a = H/2$ denotes the wave amplitude), relatively steep waves were necessary to break the ice. Nevertheless, it is important that in all tests, the wave steepness $ka < 0.05$, i.e., within the limit of the linear wave theory, so that any nonlinear effects were unlikely.

## 2.2 Floe-size data

### 2.2.1 Image processing

After five tests marked in bold in Table 1 (three in test group A and two in test group B), digital images of the ice were taken with a downward-looking crane camera. In each case, neighboring (overlapping) photographs were stitched together to obtain a single image of the broken ice sheet (see Supplementary Fig. 3 for example images). Each stitched image was subsequently processed with ImageJ and Matlab Image Processing Toolbox in order to produce a binary image of ice and water. All parameters of the algorithms were adjusted to individual images, and very bright regions present due to reflections from the lamps on the ceiling (see Supplementary Fig. 3) were corrected manually. Finally, floe boundaries were identified and each ice pixel was assigned a value corresponding to the ice floe to which it belonged. Although specialized functions of the above-mentioned software were used, several manual corrections were made at each stage, based on visual comparisons of the final results with the initial photographs (in each image, every single floe larger than $\sim 5$ cm$^2$ was inspected under strong magnification and its boundaries corrected if necessary; we estimate that for those floes, the uncertainty of the estimation of their boundaries does not exceed one pixel, i.e., it is negligible, and the relative error decreases with increasing floe size.). Figure 2 shows an example image with identified ice floes marked with different (randomly assigned) colors. Other images can be found in Supplementary Figs. 4 and 5. Table 2 provides a summary of the resulting FSD data. In each case, a total of $N_{f,\mathrm{all}}$ floes were identified. Out of this number, several very small floes, with surface areas $s < s_{\mathrm{min}} = 5$ cm$^2$, were removed before further analysis, as they represent very small pieces of ice broken off the edges of larger floes. These pieces are "small" in two ways. First, they have dimensions of just a few pixels of the original images and thus cannot be resolved properly. Second, their horizontal dimensions are comparable with the ice thickness, so that their formation is beyond the two-dimensional fracture regime analyzed here.

After tests 2060 (Group A) and 1450 (Group B), the ice sheet consisted of two regions, one much less fragmented than the other (see Supplementary Figs. 4 and 5) and therefore only those subregions were taken for further analysis, in which the crack pattern could be treated as spatially uniform. All in all, the number of floes retained for further analysis in each case equaled $N_f$ (3rd column in Table 2).

### 2.2.2 Definitions of floe-size related variables

Let us denote with $l$ a measure characterizing the linear size of the analyzed ice floes. In many studies, equivalent radius is used for $l$ (i.e., radius of a circle with surface area equal to that of the floe in question), although, obviously, the optimal choice of this measure depends on the shape of ice floes. For the discussion in this section, it is sufficient to assume that $l$ is related to the floe's basal surface area $s$ through a simple relationship $s = c_s l^2$, where $c_s$ is a constant. Further, let us denote with $n_l(l)dl$ the number of floes (within the analyzed domain) with sizes between $l$ and $l + dl$. The number-weighted floe-size distribution $p_l(l)$ can be then expressed as $p_l(l) = n_l(l)/N$, where $N = \int_0^\infty n_l(l)dl$ is the total number of floes within that domain. Analogously, let us denote with $n_s(s)ds$ the number of floes with areas between $s$ and $s + ds$. The number-weighted floe-area distribution $p_s(s)$ then is $p_s(s) = n_s(s)/N$, and we have $p_s(s)ds = p_l(l)dl$. Although area-weighted floe-size and floe-area distributions will not be analyzed in this paper, they can be obtained easily from $p_l(l)$ and $p_s(s)$ as $l^2 p_l(l)/\int_o^\infty l^2 p_l(l)dl$ and $sp_s(s)/\int_o^\infty sp_s(s)ds$, respectively.

The complementary cumulative number-weighted floe-size and floe-area distributions, describing the exceedance probability for floe sizes and areas, respectively, can be defined as: $P_l(l) = 1 - \int_0^l p_l(l')dl'$ and $P_s(s) = 1 - \int_0^s p_s(s')ds'$.

It will be shown in Section 3 that it is useful to take into account a whole set of these characteristics simultaneously, as they highlight different aspects of the analyzed data. It should be also remembered that whereas the surface areas $s$ of ice floes can be obtained directly from the digital images, other quantities characterizing the floes require certain assumptions regarding floes' shapes. In this work, each polygon representing an ice floe in the analyzed image is assigned an ellipse that has the same second central moments as the polygon. Then, for each ellipse, we determine its major axis $a_f$, minor axis $b_f$, eccentricity $e_f$, and orientation $\theta_f$. Eccentricity is defined as the ratio of the distance between the foci of the ellipse and its major axis length, so that $e_f = 0$ for a circle and $e_f = 1$ for a degenerate ellipse with $b = 0$. Orientation is defined as an angle between the tank main axis (i.e., the wave propagation direction) and the major axis of the ellipse, i.e., its absolute value varies between 0 and $90°$.

## 3  Results

### 3.1  Floe shapes and orientation

Visual inspection of Fig. 2 and Supplementary Figs. 3–5 shows that the floe shapes are far from regular. Most floes are polygonal and elongated, and they tend to be longer in the across-tank direction than in the along-tank direction. As can be seen from Fig. 3, the histograms of floes' orientation and eccentricity are similar in all five cases analyzed, with only a small fraction of floes oriented at $|\theta_f| < 45°$, i.e., with their longer axis closer to the $x$-axis than to the $y$-axis, and there are almost no floes with eccentricity $e_f < 0.5$. Moreover, importantly for the further analysis, over the whole range of values of $s$ and $b_f$ (6 and 3 orders of magnitude, respectively) there is a strong linear relationship between $b_f^2$ and $s$ (Fig. 4). Thus, $b_f$ can be regarded as a meaningful measure of the linear floe size $l$ in the wave propagation direction, and due to the fact that $b_f^2 \propto s$, we may expect $p_s(s)$ and $p_l(b_f)$ to be related through relationships described in the previous section.

## 3.2 Floe-size distributions

As already mentioned, the floe surface areas cover roughly five orders of magnitude, from $\sim 5 \cdot 10^{-4}$ m$^2$ to over 10 m$^2$. Figures 5 and 6 show the number-weighted floe-size and floe-area data from the five test runs analyzed. In Fig. 5, histograms of binned $b_f$ and $s$ values are shown in linear coordinates (with constant bin spacing); in Fig. 6 – exceedance probabilities for unbinned $b_f$ and $s$ values in logarithmic coordinates. Obviously, the histograms correspond to probability density functions $p_l$ and $p_s$, and the plots in Fig. 6 to $P_l$ and $P_s$. However, although they are just different ways of presenting the same data, it is clear that they underline certain aspects of that data and tend to obscure others. In most studies in which FSD is discussed, only log–log plots of cdfs are used, similar to those in Fig. 6 (e.g., Toyota et al., 2011, 2016; Wang et al., 2016). The shape of the curves in Fig. 6 suggests – again, similarly to data from many studies, including those cited above – the existence of two "regions", for small and large floes, with a sudden change of slope between them. Qualitatively similar shapes of $P_l(l)$ obtained from satellite and airborne floe-size data have been interpreted by the above authors as two power-law regimes. Obviously, all cumulative distributions from our tests could be fitted with two straight lines just as well. However, there are at least three important arguments against this choice. First, the "regime" of large floes covers no more than one order of magnitude in the case of $b_f$ (and, consequently, less than two orders of magnitude of $s$), which is not sufficient to speak about power-law dependence. Secondly, the histograms in Fig. 5 clearly show that in the range of medium-sized floes, roughly between 0.2 and 1.0 m in size, power law is not a good candidate distribution. Especially in tests A 2030 and 2060, the histograms have a clear maximum at $b_f \sim 0.4$ m; in the remaining three tests, no pronounced maximum exists, but nevertheless a kind of "plateau" can be observed, with values higher than a power law would imply. And thirdly, there are well established theoretical arguments against the two-power-laws concept that are relevant in the present setting – some have been mentioned in the introduction, others will be discussed in section 4 at the end of this paper.

Based on the data from our experiments, as well as insights from available research on fragmentation of brittle materials (see further Section 4), we consider the following function as a candidate for probability distribution that approximates the empirical floe-size distributions shown in Figs. 5 and 6:

$$p_l(l) = \varepsilon p_{PL}(l) + (1 - \varepsilon)p_G(l), \tag{1}$$

where:

$$p_{PL}(l) = \frac{1}{\beta^{1-\alpha}\Gamma(1-\alpha, l_m/\beta)} l^{-\alpha} e^{-l/\beta}, \tag{2}$$

$$p_G(l) = \frac{1}{\sqrt{2\pi\sigma^2}} \frac{1}{1 - \mathrm{erf}\left(\frac{l_m - \mu}{\sigma\sqrt{2}}\right)} e^{-(l-\mu)^2/2\sigma^2}, \tag{3}$$

$\alpha$, $\beta$, $\mu$, $\sigma$, and $\varepsilon$ are adjustable parameters, $\Gamma(u,x) = \int_x^\infty t^{u-1} e^{-t} dt$ is the upper incomplete gamma function, $\mathrm{erf}(x) = \frac{2}{\sqrt{\pi}} \int_0^x e^{-t^2} dt$ is the error function, and $l_m$ denotes the lowest value of $l$ for which the distributions are valid. The scaling factors in (2) and (3) ensure that $\int_{l_m}^\infty p_{PL}(l)dl = 1$ and $\int_{l_m}^\infty p_G(l)dl = 1$.

As can be seen from Eqs. (1)–(3), $p_l$ is a weighted sum of two functions: a tapered power law and a normal distribution, the relative contribution of each component dependent on the value of $\varepsilon \in [0,1]$. The power-law component has a slope $\alpha$, and the

value of $\beta$ decides on the onset of the exponential tail at large floe sizes. The second, Gaussian component of $p_l$ is significant within a limited region around $l = \mu$, with $\sigma$ describing the width of that region. The exceedance probabilities $P_{PL}(l)$ and $P_G(l)$, corresponding to $p_{PL}(l)$ and $p_G(l)$, are:

$$P_{PL}(l) = \Gamma(1-\alpha, l/\beta)/\Gamma(1-\alpha, l_m/\beta), \tag{4}$$

$$P_G(l) = \left[1 - \mathrm{erf}\left(\frac{l-\mu}{\sqrt{2}\sigma}\right)\right] \Big/ \left[1 - \mathrm{erf}\left(\frac{l_m-\mu}{\sqrt{2}\sigma}\right)\right], \tag{5}$$

and the total exceedance probability $P_l(l)$ is given by $P_l(l) = \varepsilon P_{PL}(l) + (1-\varepsilon)P_G(l)$. A detailed discussion of the properties of functions (1)–(2) and justification of their choice to represent the observed FSDs are provided in Section 4.

The distribution given by Eqs. (1)–(5) has five adjustable parameters, which makes fitting it to the data a nontrivial task, mainly due to problems with multiple local minima in the parameter space. Moreover, specific features of the pdfs analyzed here, described briefly above, make it difficult to choose a suitable approach. Methods that perform satisfactorily in terms of fitting the tails of the pdfs tend to fail in the region in the middle; and methods that successfully fit the middle parts of the pdfs fail to reproduce the tails. Nonlinear least-square fitting of $P_l(l)$ to the observed exceedance probabilities (those shown in Fig. 6) belongs to the first category – which is not surprising as even in tests 2030 and 2060, in which the maxima at floe sizes of $\sim 0.4$ m are most pronounced, hardly any signature of these maxima can be seen in cumulative distributions. Another widely used fitting method, the maximum-likelihood estimation (MLE), captures the middle regions of the pdfs, but produces tails that very strongly deviate from the observed ones. Moreover, our tests showed that both these methods are very sensitive to the value of $l_m$ (MLE is known to encounter problems with truncated distributions), as well as to the initial guess of the parameters. When ran many times with different initial conditions, both algorithms converged to very different local minima, characterized with almost identical goodness-of-fit measures, which made the choice of the "best" fit a matter of subjective preference.

Due to these problems, we tested a third approach, in which predicted cumulative probabilities are linear-least-square fitted to the empirical ones – an idea based on the fact that for a perfect fit, the cdfs should lie on a straight 1:1 line on a P–P plot (see insets in Figs. 7b,d,f and 8b,d). More precisely, the goal is to find the values of the coefficients $\varepsilon$, $\alpha$, $\beta$, $\mu$ and $\sigma$ that minimize a metrics $D$ defined as a weighted sum of the squared distances to the 1:1 line. The weights $w$ are expressed in terms of the empirical probabilities as: $w = 1/\sqrt{P_l(1-P_l)}$, i.e., they are lowest in the centre and highest at the extremes in order to compensate for the variance of the fitted probabilities, which is lowest in the tails and highest near the median. For our data, this procedure produced stable results and meaningful values of the coefficients, even though their ranges of validity had not been specified beforehand. By "meaningful values" we mean values fulfilling a few basic criteria, for example that $\varepsilon > 0$ (i.e., the contribution of both components is nonnegative) and $\mu > 0$; the two other methods often converged to $\varepsilon < 0$, $\mu \approx 0$ or $\alpha < 0$. Moreover, when tested on artificially generated data from purely Gaussian and purely power-law distributions, this method consistently produced values of $\varepsilon$ below 0.03 and above 0.97, respectively; the two methods mentioned earlier failed this test.

The values of the parameters obtained with this method are provided in Table 3. Figures 7 and 8 show the results in terms of both pdfs and cdfs for tests from Group A and B, respectively.

In order to obtain a measure of standard errors of the estimates, Monte Carlo simulations were used. For each fitted model, $N = 100$ datasets with random numbers drawn from that model were generated, the parameters were estimated by applying the procedure described above, and the standard deviation of these parameter values was used as a standard error, given in Table 3. Two-sample Kolmogorov-Smirnov tests were performed pairwise between the observed data and those generated with the fitted models. The percentage of cases in which the test rejected the null hypothesis that the two samples were from the same distribution (at the 5% significance level) varied between 2–3% for tests 2020 and 2060, 22–25% for tests 1450 and 1510, and 35% for test 2030. Additionally, the metrics $D$ was calculated for each generated model, and a $p$-value was computed defined as the percentage of cases in which $D$ was smaller than that obtained for the original data (Clauset et al., 2009; Virkar and Clauset, 2014). The lowest $p$-value was obtained for test 1450 ($p = 0.07$); the highest one for test 1510 ($p = 0.9$); all other $p$-values were close to 0.3–0.35. Thus, with the exception of test 1450 (see further), all other data can be regarded as drawn from distribution (1).

The results show that in both test groups, A and B, as fragmentation progresses, the power-law parts of the FSDs evolve towards lower values of $\alpha$ and lower values of $\beta$: the slope of the pdfs in the range of small values of $l$ decreases, and the cut-off shifts towards smaller floe sizes – which is reasonable, as less and less large floes survive without breaking. The two trends together produce larger and larger differences between the slopes of the large and small floes regions in cdf plots, giving the impression of a "regime shift". The Gaussian part of the pdfs is relatively stable, with a slight tendency for the value of $\mu$ to shift to the left, again as a result of breaking. The values if $\varepsilon$ in both tests decrease in time, indicating decreasing (increasing) contribution of $p_{PL}$ ($p_G$). Notably, in test B 1450 the predicted contribution of $p_G$ equals ∼3%, and Monte Carlo simulations produced very scattered results – note large error estimates in Table 3, especially for $\mu$ and $\sigma$. The tapered power law alone seems a more appropriate model that explains the data (last row in Table 3). Generally, the tests in Group A were conducted much longer than those of Group B (see Table 1). Group B represents early stages of fragmentation caused by relatively long waves; accordingly, the $p_l(l)$ in these tests are wider than those from Group A, and the change of slope between the region of small and large floe sizes is less pronounced. In contrast, tests 2030 and 2060 from Group B represent ice at advanced stages of breaking by short waves, in which a dominating floe size can be clearly seen in $p_l(l)$ data. Note that the Gaussian component of these pdfs contributes to the sudden change of slope in log–log cdf plots.

Note also that in the tests from Group A, the floes described by the Gaussian component of FSDs represent the "dominant" or "significant" floes in the sense that they cover the largest fraction of the total surface area, i.e., the area-weighted floe-size distributions have very peaked maxima at $b_f$ ∼0.5 m (see Supplementary Fig. 6a–c). In fact, this is also the range of values estimated by a human looking at an image of the ice like that in Fig. 2 and Supplementary Figs. 3 and 4. These maps definitely do not look "fractal". Analogous area-weighted pdfs from Test Group B, in which the power-law component is dominant, have a very different shape, with larger floes occupying larger fraction of the total surface (Supplementary Fig. 6d,e).

## 4 Discussion and conclusions

One of the conclusions of this study is that even in a simple laboratory configuration, under controlled conditions, the interpretation of the obtained floe-size distributions is far from trivial. With uniform ice, regular waves and approximately one-dimensional setting, one could expect a straightforward relationship between the wave forcing and ice mechanical properties on the one hand, and the resulting floe sizes on the other hand. However, this is not the case, and one of the main reasons for this are the properties of laboratory-grown ice, which is softer, weaker and thinner than real-world sea ice. Consequently, a number of processes contribute to breaking and overall wear out of the ice, wave-induced flexural stress being only one of the factors. Our video material clearly shows strong overwash of the upper ice surface, floe–floe collisions, grinding of small ice fragments between larger ice floes, and "erosion" of the ice producing significant amounts of slush filling spaces between ice floes at later stages of the experiments, especially those from Test Group A, in which individual runs were much longer than in Test Group B (Table 1). In runs with steeper waves (e.g., 2030, 2050 in Test Group A), several cases of floe rafting were observed as well. Importantly, the effects of these processes are visible already shortly after the formation of the first cracks, i.e., it is not possible to identify a phase of ice breaking due to flexural stresses, followed by a later phase of breaking induced by the remaining processes – they all contribute to ice fragmentation simultaneously. Consequently, although it may seem a paradox, we do not observe any regular breaking pattern similar to that repeatedly reported from the field.

Let us take a closer look at the components of function (1) in the context of what is known about fragmentation of sea ice and other brittle materials. The function postulates that the observed floe-size distributions are a result of two (groups of) processes, one leading to scale invariance of floe sizes, with some tapering effects present at large floe sizes, and the other producing a preferred floe size, with some random scatter around the mean value. A similar general approach, in which the probability distribution of fragment sizes is expressed as a sum of two (or more) terms, is well known in studies on fracture of brittle materials. Multimodal distributions observed in some fragmentation experiments are often fitted with bilinear Poisson distributions, with individual components attributed to distinct fracture mechanisms significant at distinct spatial scales (see, e.g., Grady, 2006). One interesting example, relevant in the present context, is breaking of slender, elongated rods made of a brittle material, as, e.g., in the experiments of Gladden et al. (2005), in which rods made of dry pasta, glas, steel, and so on, impacted axially, undergo a dynamic buckling instability and break. The resulting fragment-length distributions are nonmonotonic, i.e., they exhibit maxima corresponding to the dominating wavelengths of the perturbation developing in the material shortly before the onset of breaking (see Fig. 5 in Gladden et al., 2005). As the authors note, the effects of fragmentation in this case are not purely random, but "include the imprint of the deterministic buckling process leading to breakup". Higley and Belmonte (2008) referred to this mode of fragmentation as "patterned breaking" and proposed a one-dimensional mathematical model of this fragmentation mechanism, in which the probability density of breaking is a prescribed function of location. The model successfully predicted the observed distributions of fragment sizes. Crucially, although stress maxima corresponding to the locations of maximum curvature of the rod are regularly distributed along its length, the observed fragment-size distributions are very wide, due to a number of competing effects acting in parallel, including flexural waves associated with stress release after individual breaking events, pre-existing flaws in the material or so-called delayed-fracture phenomenon (Vandenberghe

and Villermaux, 2013). In a different context, Åström et al. (2014) analyzed observed and simulated calving rates at grounded tidewater glaciers and floating ice shelves. They showed that fragment-size distributions obtained from their data can be described as a sum of two components, one representing the largest fragments and dependent on the large-scale pattern of parent cracks, and the other resulting from crack propagation and grinding within individual fracture zones. Riikilä et al. (2015) used the same approach in their discrete-element model of glacier ice and analyzed how model parameters influenced the relative contribution of the two components to the resulting fragment-size distributions.

Analogously to the studies mentioned above, it seems reasonable to represent the floe-size data with a function given by Eq. (1), with one component describing the "patterned breaking" due to wave-induced flexural stress, acting at a clearly defined spatial scale, and the other component representing the remaining fracture mechanisms, producing floes with sizes spanning a few orders of magnitude. As has been mentioned in the introduction, the recent numerical studies on ice breaking by waves suggest that the Gaussian distribution $p_G(l)$ is a suitable candidate for the first component. For an ice sheet floating on the water "foundation" and subject to flexural deformation, the location of the maximum bending stress relative to the ice edge – and thus the most probable breaking location – can be estimated from:

$$x_m = \frac{\pi}{2} \left( \frac{E h_{\mathrm{ice}}^3}{3 k_w (1 - \nu^2)} \right)^{1/4}, \tag{6}$$

where $k_w$ is the foundation (in this case: water) modulus and $\nu$ is the Poisson's ratio (see, e.g., Mellor, 1986). For $k_w = 10^4$ N·m$^{-3}$, $\nu = 0.3$ and the values of $E$ and $h_{\mathrm{ice}}$ measured in our experiments (see Sections 2.1.2 and 2.1.3), we obtain $x_m = 0.48$ m for Test Group A and $x_m = 0.62$ m (based on the average ice thickness) for Test Group B. Remarkably, this is very close to the values of $\mu$ obtained during the fitting process (Table 3), especially for the first group of tests, in which, as we describe in Section 2.1.2, breaking progressed gradually from the ice edge, so that the assumptions underlying (6) should be valid. This is in agreement with Squire et al. (1995) and with the recent results by Herman (2017) showing that the floe size resulting from breaking by waves depends not on the incoming wavelength, but rather on the mechanical properties of the ice itself.

The second component of (1) is more problematic, as its suitable form depends on the character of the fragmentation process. Fragment-size distributions in the form of a power law with an exponential cut-off, as given by $p_{PL}(l)$, have been reported in numerous studies of fragmentation in both two and three dimensions (including those by Åström et al., 2014; Riikilä et al., 2015, cited above), and models explaining the emergence of power-law fragment size distributions have a sound theoretical basis (see, e.g., Kekäläinen et al., 2007; Åström et al., 2000, 2004). In these models, the power-law "regime" of fragment sizes results from branching and merging of cracks produced around major, parent cracks, and as the energy available for new crack production is limited, the width of the fracture zone and thus the fragment size is limited as well, producing the exponential cut-off in the observed probability distributions. Thus, the cut-off results from the nature of the process itself. Another source of a cut off are finite-size effects that obviously are significant or even dominating in many configurations. Undoubtedly, in a laboratory experiment the floe sizes are subject to a global constraint $\sum_i s_i = S_{\mathrm{tot}}$, where $S_{\mathrm{tot}}$ denotes the surface area of the ice sheet. The influence of global constraints of this kind on the tails of power-law pdfs is discussed in Sornette (2006). Together with waves acting as a floe-size limiting factor, this eliminates the possibility of obtaining FSDs with heavy, power-

law tails. As it is well documented that the Gamma distribution is found in critical phenomena in the presence of finite size effects, this functions seems suitable for representing FSD data. Notably, Gherardi and Lagomarsino (2015) use a very similar functional form – a power law with an exponential cut-off – to describe the observed FSD data from four different regions. More importantly, they analyze two different statistical models of fragmentation, both of which are shown to produce power laws with exponential cut-offs. Notably as well, Lu et al. (2008) used the Weibull distribution (i.e., a pdf in the form of a product of a power-law term and an exponential term) to fit their observational FSD data.

Importantly, branching models of fragmentation predict that the exponent of the power law is universal and depends only on the dimension $D$ in which the process takes place: $\alpha_D = (2D-1)/D$ (Åström et al., 2004; Kekäläinen et al., 2007). In two dimensions, relevant for sea ice breaking at scales larger than ice thickness, this value relates to pdf of surface areas, $p_s(s)$: $\alpha_s = \alpha/2 = 3/2$. Values of $\alpha_s > 3/2$ are expected in situations when fragmentation due to crack propagation is accompanied by further grinding of the material under combined compressive and shear deformation (e.g., Oron and Herrmann, 2000). Scale-invariance in these models and observations suggests that fragmentation takes place as a self-organized process, as opposed to random breaking that results in exponential fragment-size distributions (e.g., Grady, 2006), i.e., $\alpha_s = 0$.

In the experiments described here, $\alpha$ was close to 1 during initial tests (2020 in Group A and 1450 in Group B) and decreased to values as low as 0.24 in test 2060. This suggests, reasonably, that the random breaking model is more appropriate in this case. The video material collected during the experiments shows that individual cracks seem to form independently of each other, have simple, linear form, i.e., without secondary, rapidly forming side branches. To the contrary, formation of individual cracks is relatively stretched in time – it begins at the lower side of the ice sheet and may take a few wave periods until the two new ice floes detach from each other. This behavior is very different from processes that are described by the branching models, in which crack formation is rapid and their dynamic instability is the source of branching and the resulting scale-invariance of fragment sizes. It must be also remembered that the ice floes in our experiments were allowed to drift towards the open water area in front of the wavemaker, so that the conditions were very far from those favorable for grinding. High values of $\alpha$ are rather expected under confined conditions dominated by compressive, not tensile deformation.

Finally, it is worth noting that the processes that lead to breaking of the ice influence not only the sizes, but also the shape of the ice floes. As has been noted in Section 3.1 (see also Fig. 2 and Supplementary Figs. 3–5), the floes obtained in the experiments described here were polygonal, with relatively straight edges and sharp angles. Similar (or even more regular, rectangular) floe shapes have been observed in sea ice broken by waves (e.g., Squire, 1984; Langhorne et al., 1998; Squire and Montiel, 2015). They are very different from approximately circular floes often observed in satellite images. In the literature, floe shapes attracted much less attention than the floe-size distribution (but see Gherardi and Lagomarsino, 2015) and little is known about factors influencing their evolution, but it is tempting to speculate that in an initially intact ice sheet broken by waves, angular floes are formed that subsequently gradually evolve towards more rounded shapes (and wider size distributions) in a process of grinding, known to produce rounded grains in other granular materials.

In general, the results presented here, obtained under controlled laboratory conditions, illustrate how difficult is the interpretation of real-world floe-size data – when the ice floes are a product of many cycles of breaking, freezing, melting and so on. In most cases, only snapshots of the ice cover are available, without information on its history and forcing acting on it. Nev-

ertheless, we believe that more insight could be gained from the existing FSD data sets. It would be worthwhile to reexamine the published floe-size data without commonly made *a priori* assumptions regarding the form of the pdfs and to test alternative floe-size distribution models.

*Author contributions.* K.-U.E. and N.R. planned and conducted experiments in Test Group A. All authors planned and conducted experiments in Test Group B and contributed to the discussion of the results. A.H. performed the data analysis and wrote the manuscript.

*Acknowledgements.* The work described in this publication was supported by the European Community's Horizon 2020 Research and Innovation Programme through the grant to HYDRALAB-PLUS, Contract no. 654110. The authors would like to thank the Hamburg Ship Model Basin (HSVA), especially the ice tank crew, for the hospitality, technical and scientific support and the professional execution of the test programme in the Research Infrastructure ARCTECLAB. We are also very grateful to our colleagues from the LS-WICE project: Andrei Tsarau, Hayley Shen, Hongtao Li, Sveinung Løset and Sergiy Sukhorukov, for their cooperation and support. We thank Chris Horvat and an anonymous Reviewer for their insightful comments on the first version of the manuscript.

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

**Table 1.** Summary of test runs discussed in this paper

| Run No. | Wave height $H$ (cm) | Wave period $T$ (s) | Wave length $L$ (m) | $kh_{ice}$ (–) | Test duration $t_w$ (min) | Remarks |
|---|---|---|---|---|---|---|
| Test Group A | | | | | | |
| 2010 | 2.0 | 1.27 | 2.52 | 0.075 | 10 | no breaking observed |
| **2020** | 5.0 | 1.27 | 2.52 | 0.075 | 10 | breakup up to $x \sim 34$ m |
| **2030** | 7.0 | 1.27 | 2.52 | 0.075 | 10 | breakup only in already broken zone |
| 2050 | 10.0 | 1.27 | 2.52 | 0.075 | 10 | a few new cracks for $x > 34$ m |
| **2060** | 10.0 | 1.50 | 3.51 | 0.054 | 11 | breakup of the whole ice sheet |
| Test Group B | | | | | | |
| 1100 | 1.0 | 2.0 | 6.17 | 0.035 | 1.5 | no breaking observed |
| 1200 | 1.0 | 1.6 | 3.99 | 0.055 | 1.5 | no breaking observed |
| 1300 | 1.0 | 1.2 | 2.25 | 0.097 | 1.5 | no breaking observed |
| 1400 | 2.0 | 2.0 | 6.17 | 0.035 | 1.5 | no breaking observed |
| 1410 | 3.0 | 2.0 | 6.17 | 0.035 | 1.5 | no breaking observed |
| 1420 | 4.0 | 2.0 | 6.17 | 0.035 | 1.5 | no breaking observed |
| 1430 | 5.0 | 2.0 | 6.17 | 0.035 | 1.5 | no breaking observed |
| 1440 | 7.0 | 2.0 | 6.17 | 0.035 | 2.0 | first major crack at $x \sim 44$ m |
| **1450** | 9.0 | 2.0 | 6.17 | 0.035 | 3.5 | major breakup, esp. downwave of $x \sim 44$ m |
| 1500 | 5.0 | 1.6 | 3.99 | 0.055 | 1.8 | only a few new cracks |
| **1510** | 7.0 | 1.6 | 3.99 | 0.055 | 6.2 | major breakup of the whole ice sheet |

Tests after which photos of the ice were taken are shown in bold. Tests 1450 and 1510 were continued until no breaking occured.

Virkar, Y. and Clauset, A.: Power-law distributions in binned empirical data, Ann. Appl. Statistics, 8, 89–119, doi:10.1214/13-AOAS710, 2014.

Wang, Y., Holt, B., Rogers, W., Thomson, J., and Shen, H.: Wind and wave influences on sea ice floe size and leads in the Beaufort and Chukchi Seas during the summer-fall transition 2014, J. Geophys. Res., 121, doi:10.1002/2015JC011349, 2016.

5   Williams, T., Bennetts, L., Squire, V., Dumont, D., and Bertino, L.: Wave-ice interactions in the marginal ice zone. Part 1: Theoretical foundations, Ocean Modelling, 71, 81–91, doi:10.1016/j.ocemod.2013.05.010, 2013.

Williams, T., Rampal, P., and Bouillon, S.: Wave–ice interactions in the neXtSIM sea-ice model, The Cryosphere, 11, 2117–2135, doi:10.5194/tc-11-2117-2017, 2017.

Zhang, J., Schweiger, A., Steele, M., and Stern, H.: Sea ice floe size distribution in the marginal ice zone: Theory and numerical experiments, 10   J. Geophys. Res., 120, 1–15, doi:10.1002/2015JC010770, 2015.

**Table 2.** Summary of FSD data obtained from the analyzed images

| Run No. | No. of floes | | Mean area | Median area |
|---------|-------------|-------------|-------------|-------------|
| | $N_{f,\text{all}}$ | $N_f$ | $s_{\text{mean}}$ (m$^2$) | $s_{\text{med}}$ (m$^2$) |
| A 2020 | 1683 | 705 | 0.25 | 0.01 |
| A 2030 | 1605 | 1036 | 0.16 | 0.05 |
| A 2060* | 1017 | 777 | 0.19 | 0.08 |
| B 1450* | 1508 | 814 | 0.66 | 0.01 |
| B 1510 | 1779 | 848 | 0.53 | 0.01 |

In tests marked with a star, a subregion of the whole ice sheet was analyzed, in which FSD could be treated as spatially uniform. The numbers in the table correspond to these subregions. $N_{f,\text{all}}$ – No. of all floes identified; $N_f$ – No. of floes used in the analysis (see Section 2.2.1).

**Table 3.** Results of least-square fit of Eq. (1) to observed FSD data

| Run No. | $\varepsilon$ | $\alpha$ | $\beta$ | $\mu$ | $\sigma$ |
|---------|--------------|----------|---------|-------|----------|
| A 2020 | 0.821±0.119 | 1.039±0.250 | 0.736±0.271 | 0.574±0.039 | 0.160±0.060 |
| A 2030 | 0.685±0.039 | 0.590±0.084 | 0.298±0.039 | 0.431±0.012 | 0.111±0.014 |
| A 2060 | 0.610±0.068 | 0.245±0.115 | 0.204±0.037 | 0.463±0.022 | 0.154±0.021 |
| B 1450 | 0.968±0.042 | 1.136±0.115 | 2.408±0.676 | 1.117±3.280 | 0.055±1.265 |
| B 1510 | 0.695±0.030 | 0.513±0.169 | 0.155±0.035 | 0.924±0.053 | 0.391±0.037 |
| B 1450 | $\varepsilon = 1$ | 1.123±0.065 | 2.743±1.776 | — | — |

The error estimates are standard deviations obtained with Monte-Carlo simulations (see text). The last row shows LS fit of data from test 1450 to a tapered power law ($\varepsilon = 1$).

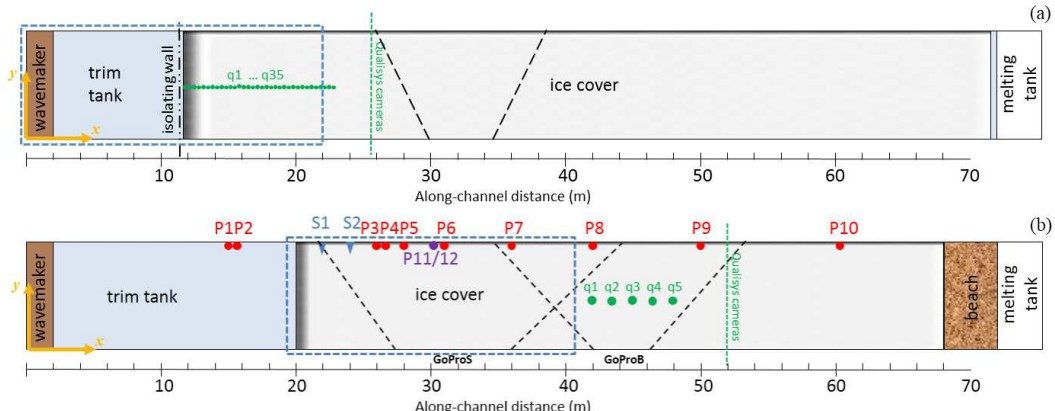

**Figure 1.** Instrument setup during Test Group A (a) and B (b): single pressure sensors are marked in red, a double pressure sensor – in violet, ultrasound sensors – in blue, Qualisys markers – in green; dashed black lines show fields of view of sideward-looking GoPro cameras, dashed blue lines – fields of view of the cameras mounted on the ceiling.

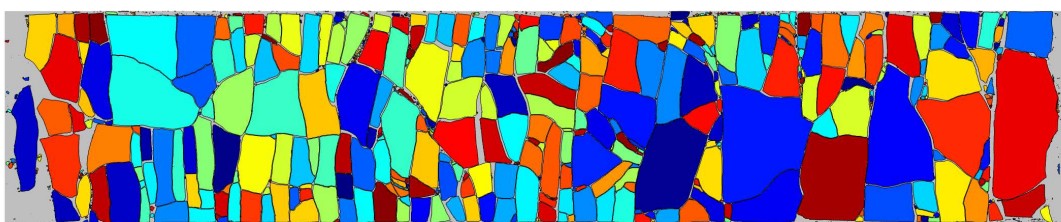

**Figure 2.** Example of a final result of the image analysis (test B, 1510), with identified ice floes marked by black contours and randomly selected colors. The ice edge is to the left, the beach to the right; the height of the image corresponds to the distance of 10 m (tank width), gray areas are open water or ice that could not be identified (very small pieces etc.). See Supplementary Figs. 2 and 3 for all analyzed images.

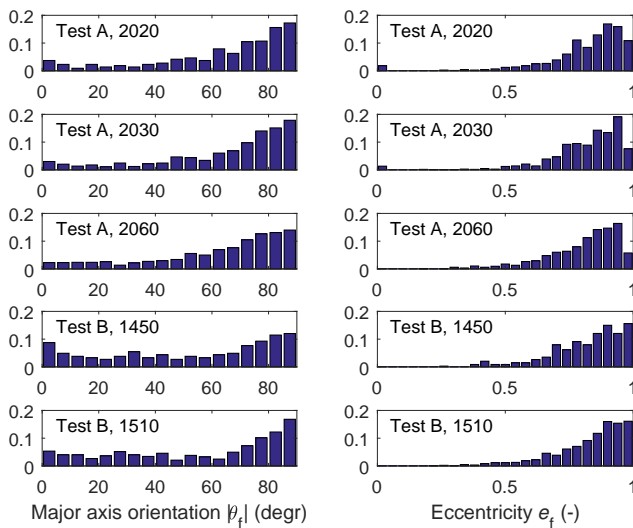

**Figure 3.** Histograms of the absolute value of floe orientation $|\theta_f|$ (left) and eccentricity $e_f$ (right) in the five analyzed tests, for floes with $s > 5 \cdot 10^{-3}\text{m}^2$. Bin widths equal 5° and 0.04, respectively; bar heights are normalized so that their total area in each panel equals one.

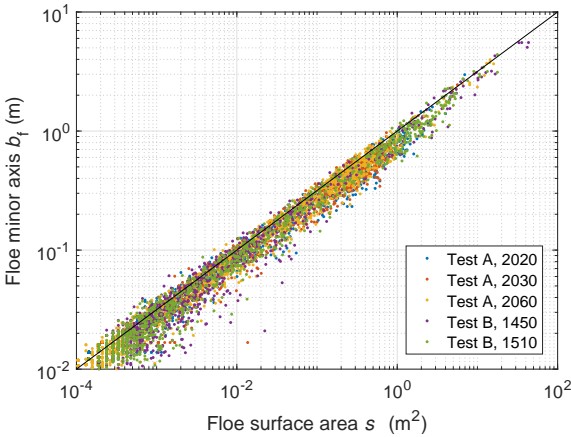

**Figure 4.** Scatterplot of the floe minor axis $b_f$ (m) *vs.* floe surface area $s$ (m²). Data from all five tests. The slope of the black line equals 2.

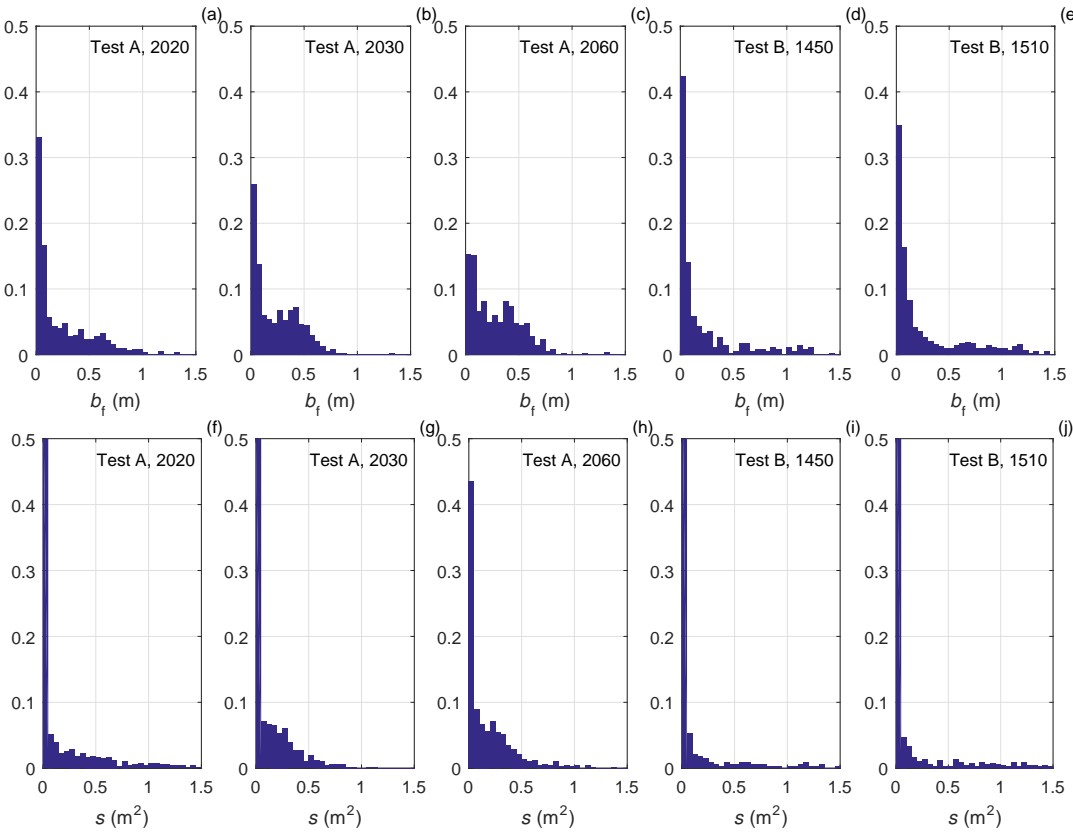

**Figure 5.** Histograms of $b_f$ (a–e) and $s$ (f–j) from all five tests analyzed. Bin width equals 0.05 m and 0.05 m$^2$, respectively.

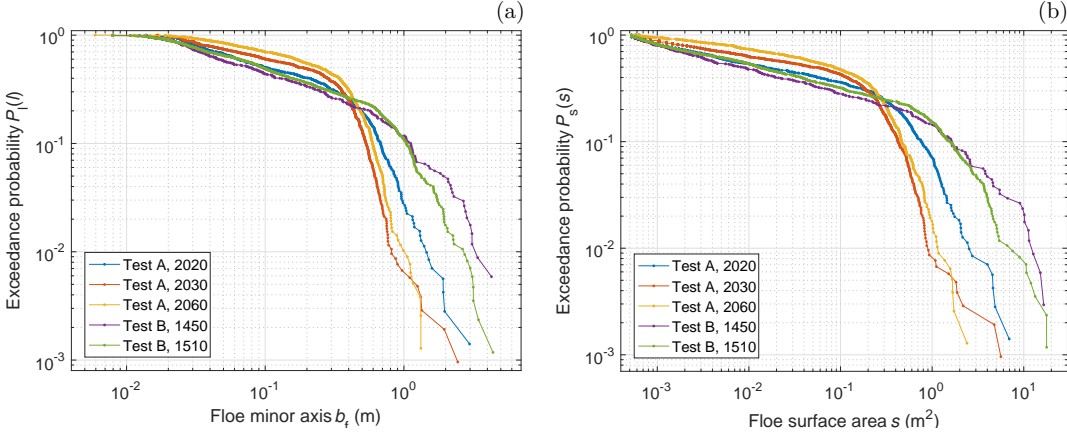

**Figure 6.** Log–log plots of the exceedance probability $P_l(l)$ (a) and $P_s(s)$ (b) for unbinned data from all five tests, for floes larger than $5 \cdot 10^{-4}$ m$^2$.

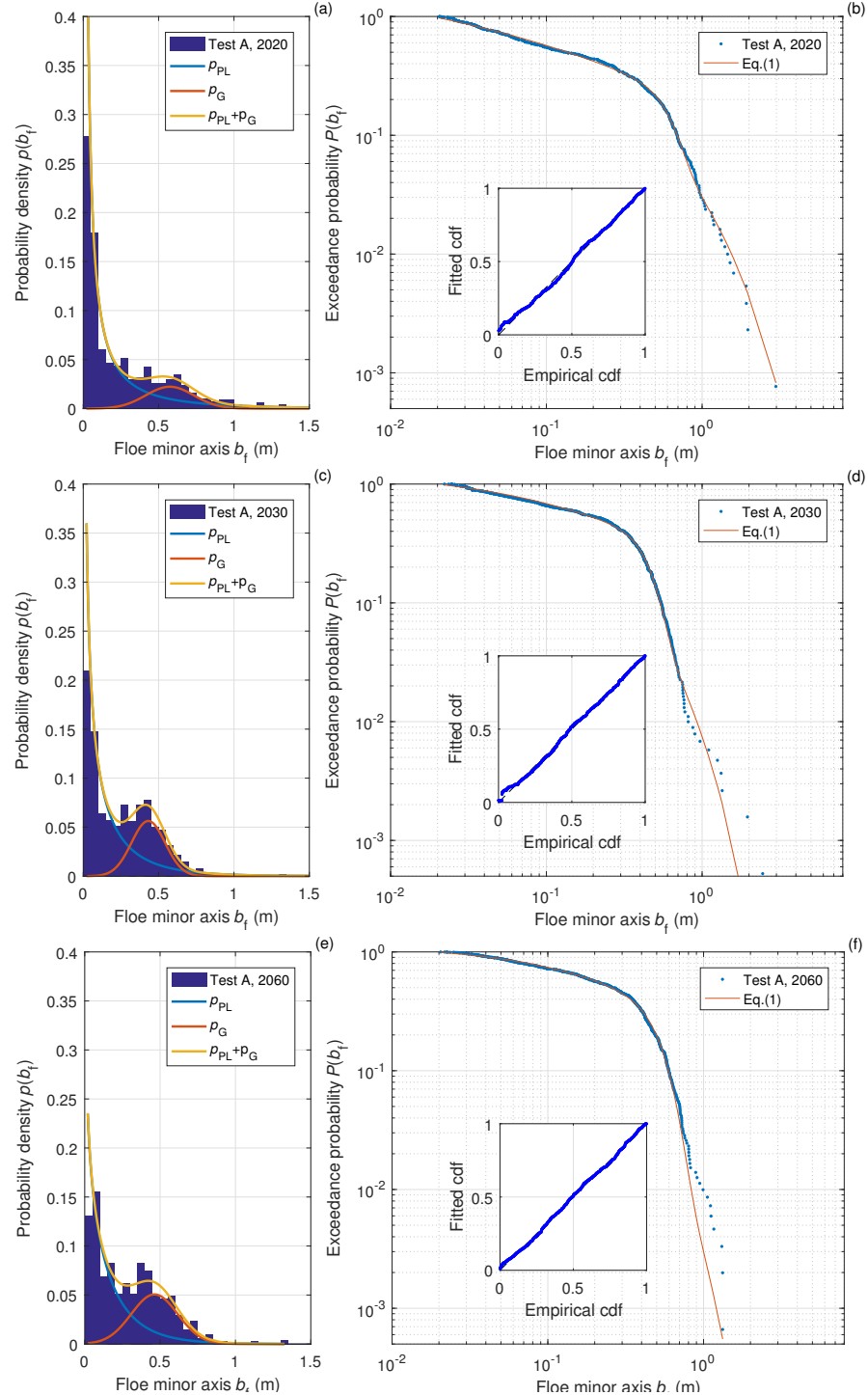

**Figure 7.** Results of the linear least-square fit of predicted and observed cdfs for $b_f$ data from Test Group A: histograms of $b_f$ with fitted $p_{PL}$, $p_G$ and $p_l$ (a,c,e) and observed exceedance probabilities with fitted $P_l$ (b,d,f). The insets show P–P plots of the fitted *vs.* observed cdfs.

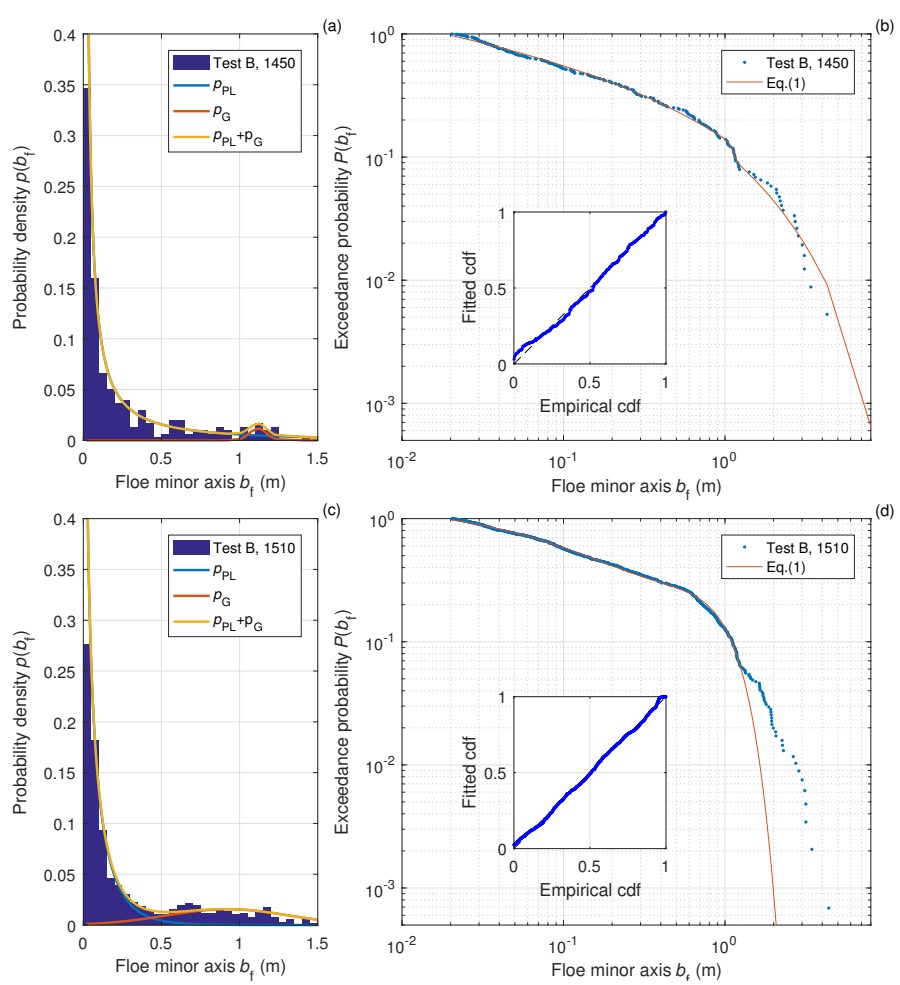

**Figure 8.** As in Fig. 7, but for Test Group B.