# Peer review of "Floe-size distributions in laboratory ice broken by waves"

_The Cryosphere, 2017_

## Referee Comment (RC1) · C. Horvat (Referee) · 20 Nov 2017

This manuscript concerns laboratory experiments in which lab-grown ice is broken by water surface waves, and the shape and type of ice pieces formed due to the fracture process. The laboratory experiments are clearly articulated and the rationale is quite clear. The authors do a good job of discussing an important area of sea ice physics that will benefit greatly from such experimental work. I have concerns about the fitting procedure and analysis that should be cleaned up in review, but otherwise find this manuscript to be important work worthy of publication. Comments are below.

Introduction.

Page 1 Line 23 - Please cite some of the "growing evidence that the FSD... " etc as

this is not supported in the text.

Page 2 Line 10 - Here some discussion of Virkar and Clauset (2012), and subsequent papers would be useful.

Page 2 Line 17 - I would be more careful to avoid being subject to the same criticisms you levy earlier: "obtained Gaussian pdfs" (the claim) and "fit observations to a Gaussian" (the reality) are different things. Same with the later comment on the FSDs produced using DESIgn.

Experimental Design

I'm a little unclear as to the applicability of these test conditions to the real world. The scale of fractured ice and surface waves in the model experiment are quite small relative to the scale of floes even in the Southern Ocean. Obviously there are experimental constraints but a discussion of how this 72x10x2.5/5 m tank relates to a strip of ice on the sea ice margin would be helpful. In addition, where might one expect the thermodynamic conditions?

Page 3 Line 17 - If I read this correctly, there are really only two tests being performed rather than many groups of tests, as one sheet of ice is broken. Does one expect path-independence of the FSD? If not this is a shortcoming that should be discussed.

Page 3 Line 30 - What about salinity/salt in the water?

Page 4 Line 23 - How do these attenuation rates relate to those used in popular attenuation parameterizations?

Page 5 Line 1 - Again, how does this relate to sea ice conditions in the "real world"?

Page 5 - Image processing - Can you be more specific about the image processing methodology? For example, to produce a binary image one might employ a thresholding value, but the results may be very sensitive to this parameter. How substantial were the "manual corrections", and how sensitive were the final results to the image process

parameters?

Page 6 Line 5 - I think one should re-define "surface area" as "basal surface area", or simply "area" here, as much of the interest in the FSD has focused on the "lateral surface area" component of floes, which seems to impact lateral melting (i.e. Steele, 1992, Roach et al, 2017).

Page 6 Line 10 - Some discussion of what the "number-weighted FSD" is would be helpful, as it is not clear from this how the distributions discussed here are related to the other "FSD"s that proliferate in the literature.

Page 7 Line 6 - If interest is in the FSD, why should we are about the range of order of magnitudes of surface area, why not report this in terms of the distribution of effective radii, or at least b_f?

Page 8 Line 1 - The entire discussion on fitting is somewhat difficult to parse given the authors (correct) insistence on mathematical and scientific scrutiny of the power-law hypothesis. The 5 adjustable parameters have little connection to physics, and the concept of what "meaningful values" of the tunable parameters might be is unclear. While the authors do some testing of the coefficients, they don't do any real hypothesis testing. A way of doing this is to draw random distributions from the model, and compute a p-value based on the fraction of those random distributions are closer to the model than the observed distribution (via a K-S statistics, for example).

This is not the only hypothesis worth testing: another is that any data would pass this test. One could perform the same test, but replace the observed distribution with a power-law or gaussian. Given the 5 adjustable parameters, I think this needs to be done.

Page 9 - Discussion

How does one explain the relatively rectangular character of the ice here, relative to the relatively circular character of the ice in real conditions? Is the grinding of floes a

significant factor in this?

I would like to see a plot of something like mean floe size in time (or as a function of breaking event #), as this is of importance for models of the FSD.

The discussion of the sum-of-two-distributions idea is well-taken, but a discussion of other processes that act on floes other than those influenced by waves would be good to have, in particular the fact that these processes may or may not be dominant region-ally or hemispherically (i.e. for example, if in the Southern Ocean, waves are important but not in the Arctic).

The authors are free to contact me with any questions.

---

## Referee Comment (RC2) · Anonymous Referee #2 · 30 Nov 2017

General comments: This paper aims at improving the understanding of the FSD observed so far by examining the properties of the floe size distribution (FSD) produced by wave-induced fracturing through laboratory tank experiments. The major purpose of this study is to show what kind of FSD is formed under as simplistic a situation as possible with regular waves and thin uniform ice, and then to check the applicability of a power-law function which has widely been used. Resultantly, they found that even in such a simple configuration, the FSD produced is far from simple. Based on the results, they proposed that a summation of two functions, the upper incomplete gamma function and the normal distribution, would represent the real FSD rather than two regimes of power-law functions with different exponents. Overall, I feel that this paper is a good and interesting work, and this approach is indispensable to improve our understanding

of FSD of sea ice. Therefore, I believe this work will contribute to the development of sea ice dynamics, especially for the formation processes of FSD, related to the wave-ice interaction. Even so, I consider this paper needs somewhat more insight about physical interpretation of the results. My major comments are as follows:

1) Regarding the fitting functions, could you explain why you consider these two types of functions (the gamma function and the normal distribution) are appropriate for representing the wave-induced FSD? I could understand from the description in Discussion section that there seems to be two different processes and it would be appropriate to represent the FSD by a summation of two different functions. I can understand one should be the normal distribution if there is a representative length. But how do you explain physically about the gamma function? Since this issue seems to be the key of this paper, it might be better to add some brief explanation earlier, e.g. when the two functions were defined in section 3.2. 2) Besides, how do you explain the representative size of the normal distribution? The value of about 0.5 m observed in the tank is clearly different from the wavelength. As the value of 0.5 m is almost common for all the experiments, it might be related to the ice property such as minimum buckling size determined by the stiffness of the material (Mellor, 1986). Please try to find a reasonable explanation. I think this is important to apply the result to the real FSD in sea ice. 3) Regarding the description "scale-invariance of floe size distribution is assumed a priori" (P2L3), and "In many cases, no convincing arguments for assuming power-law FSDs exist" (P2L7-8), I do not agree. Although I might be wrong, to my understanding, this is not necessarily true. I understand researchers did not assume scale-invariance a priori, but examined it by showing how well a power-law function fits the FSD observed. Personally, I consider there is a good reason for thinking of a power-law as a candidate of PDF function. In nature, it would be natural that size distribution (for any material) tends to become scale invariant without any external forcing that determines the scale. Historically, it has been well known that there is a scale invariant property in the process of sea ice breakup (Weiss, 2001 Engineering Fracture Mechanics Vol.68). In this experiment, a scale given by external forcing would be wavelength. Therefore,
it might be possible that the FSD for floes smaller than the wavelength becomes scale invariant through the breakup process and follows a power-law function. Also in your results, there is a possibility that floe size may have a scale invariance property, judging from Fig.2, although you asserted "do not look fractal" in P9L22. (For example, I recommend you plot long-axis against short-axis of the ellipse for individual floes. It might show almost linear relationship.)

Besides, it should be kept in mind that in reality the FSD formation process is not limited to the wave-induced breakup of sea ice. The herding and other processes that do not have specific scales, which is induced by winds and/or currents, can also contribute to the FSD formation as pointed out by Toyota et al. (2011). Therefore, when the controlling scale is unclear, I do not think it is unreasonable to try to use a power-law function for fitting to test the scale invariance. What do you think? (But as far as this experiment is concerned, I agree that the normal distribution should be used because regular waves with a fixed wavelength can determine the dominant scale.) Thus, I recommend you reconsider about this matter.

Specific comments: *(P2L8-9) "no alternative pdfs are considered" This is not necessarily true. For example, Lu et al. (2008) used the Weibull distribution for fitting. *(Section 2) Please describe whether you used pure ice or sea ice in the experiment. It would provide useful information to consider the ice flexural strength. *(Table 1) How did you obtain the wavelength? Was it measured directly or estimated from the theory of deep water approximation? *(Table 2) Please add the explanation of Nf and Nf,all in the caption rather than in the text. *(Section 4) It would be desirable if you include some description about how these results can be applied to the real FSD of sea ice.

That is all. Faithfully yours.

---

## Author Comment (AC1) · 10 Jan 2018

We would like to thank both Reviewers for their insightful comments and criticism. Even though, as we write below, we don't always agree with the comments, they made us reconsider some important issues and modify the manuscript accordingly. We believe its quality has improved thanks to that.

**REPLY TO THE COMMENTS OF REVIEWER No. 1 (C. Horvat)**

*1. Page 1 Line 23 - Please cite some of the "growing evidence that the FSD..." etc as this is not supported in the text.*

We added some references to this statement, with examples of papers in which the

influence of dynamic and/or thermodynamic processes on the evolution of the FSD shape is dicussed.

*2. Page 2 Line 10 - Here some discussion of Virkar and Clauset (2012), and subsequent papers would be useful.*

Following this suggestion, we added references to Clauset et al., 2009, and Virkar and Clauset, 2012, with a short list of typical problems related to linear regression to log–log plots.

*3. Page 2 Line 17 - I would be more careful to avoid being subject to the same criticisms you levy earlier: "obtained Gaussian pdfs" (the claim) and "fit observations to a Gaussian" (the reality) are different things. Same with the later comment on the FSDs produced using DESIgn.*

Yes, this is true. In the revised manuscript we cite the expression that Montiel and Squire used to describe their pdfs: "nearly normal" (!).

*4. I'm a little unclear as to the applicability of these test conditions to the real world. The scale of fractured ice and surface waves in the model experiment are quite small relative to the scale of floes even in the Southern Ocean. Obviously there are experimental constraints but a discussion of how this 72x10x2.5/5 m tank relates to a strip of ice on the sea ice margin would be helpful. In addition, where might one expect the thermodynamic conditions?*

Obviously, the purpose of laboratory measurements is not to reproduce "real-world" conditions 1 to 1. In this case, what is important is not that the ice sheet in the lab was "small", but rather its dimensions relative to the dimensions of the waves. We agree that we should have made some comments on that in our manuscript. In the revised version, we added a new subsection 2.1.4, in which we provide values of the $kh_{\text{ice}}$ etc. used in the tests and relate them to the corresponding field conditions.

As for the thermodynamic conditions: as we write in subsection 2.1.1, all tests were
made at temperatures close to 0°C in order to avoid freezing of the ice sheet to the side walls of the tank, as well as ice formation on our instruments. Therefore, the experiments definitely are not representative for conditions at very low temperatures, when rapid freezing and frazil formation takes place between ice floes and may influence wave propagation. At the same time, sea ice breaking by waves (that is, single breaking events) is a very rapid process, with time scales involved that are much shorter than those typically associated with freezing/melting and other thermodynamic processes. It seems reasonable to assume that in the context of our experiments, the only really important factor related to thermodynamics is the temperature through its strong influence on the elastic modulus and, to a less extent, flexural strength of the ice.

*5. Page 3 Line 17 - If I read this correctly, there are really only two tests being performed rather than many groups of tests, as one sheet of ice is broken. Does one expect path-independence of the FSD? If not this is a shortcoming that should be discussed.*

No, no path-independence is expected (as our discussion section, in which we stress that the FSD is a combined effect of many different processes, should make clear). We added a sentence to this paragraph to make it absolutely clear that there was only one ice sheet per test group.

*6. Page 3 Line 30 - What about salinity/salt in the water?*

We added the information on the salinity of water and ice to subsection 2.1.1.

*7. Page 4 Line 23 - How do these attenuation rates relate to those used in popular attenuation parameterizations?*

The range of values of the attenuation rates that can be found in the literature is very wide, and the dependence of the attenuation rate on the wave period and ice type is far from established (e.g., Cheng et al., 2017). When the attenuation rates from our experiments are expressed per wavelength instead of per meter, they are well within the range of observations. We provide that information in the new Section 2.1.4.

*8. Page 5 Line 1 - Again, how does this relate to sea ice conditions in the "real world"?*

See the new Section 2.1.4.

*9. Page 5 - Image processing - Can you be more specific about the image processing methodology? For example, to produce a binary image one might employ a threshold-ing value, but the results may be very sensitive to this parameter. How substantial were the "manual corrections", and how sensitive were the final results to the image process parameters?*

As can be seen from example images in Supplementary Fig. 3, producing binary images from the color ones was not trivial due to reflections from the lamps on the ceiling and differences in brightness of individual images from which the composite image was assembled. Due to the fact that the total number of cases equaled only 5, it was possible, first, to adjust all parameters of the algorithms to individual images, and second, to verify the final result against the original image for almost every single floe larger than $\sim$5cm$^2$, and to manually correct e.g. boundaries between floes touching each other and not recognized by the algorithm. Each fragment of each image was carefully and painstakingly inspected under strong magnification. We estimate that for floes larger than 5cm$^2$, considered in our analysis, the errors are negligible, as they do not exceed an area corresponding to a one-pixel-wide strip around the floe perimeter (which means also that the relative errors get smaller for larger floes). We added some information about that to the revised Section 2.2.1.

*10. Page 6 Line 5 - I think one should re-define "surface area" as "basal surface area", or simply "area" here, as much of the interest in the FSD has focused on the "lateral surface area" component of floes, which seems to impact lateral melting (i.e. Steele, 1992, Roach et al, 2017).*

We changed it to "basal surface area", as suggested.

*11. Page 6 Line 10 - Some discussion of what the "number-weighted FSD" is would be*

*helpful, as it is not clear from this how the distributions discussed here are related to the other "FSD"s that proliferate in the literature.*

But this paragraph contains all relevant definitions! The number-weighted FSDs (especially in terms of cdfs, not pdfs) are the ones most frequently used in the literature.

*12. Page 7 Line 6 - If interest is in the FSD, why should we are about the range of order of magnitudes of surface area, why not report this in terms of the distribution of effective radii, or at least $b_f$?*

In the paragraph that begins with this sentence we write about both $b_f$ and $s$, and both quantities are shown in Figs. 5 and 6 that are referenced there. Moreover, the criterion for the minimum floe size taken into account in the analysis was formulated based on $s$, not $b_f$, so it seemed natural for us to write it this way.

*13. Page 8 Line 1 - The entire discussion on fitting is somewhat difficult to parse given the authors (correct) insistence on mathematical and scientific scrutiny of the power-law hypothesis. The 5 adjustable parameters have little connection to physics, and the concept of what "meaningful values" of the tunable parameters might be is unclear. While the authors do some testing of the coefficients, they don't do any real hypothesis testing. A way of doing this is to draw random distributions from the model, and compute a p-value based on the fraction of those random distributions are closer to the model than the observed distribution (via a K-S statistics, for example). This is not the only hypothesis worth testing: another is that any data would pass this test. One could perform the same test, but replace the observed distribution with a power-law or gaussian. Given the 5 adjustable parameters, I think this needs to be done.*

By "meaningful" values we only mean those that fulfill some very basic criteria; for example, it seems reasonable to expect that the mean of the Gaussian part of the distribution should be positive. Or that $\varepsilon$ should be positive if it is meant to describe the relative contribution of the two "basic" pdfs. We do not prescribe any other criteria that the "acceptable" fits should fulfill. As we write in the text: the range of validity of the

coefficients was not specified during the fitting process. We rewrote this comment in the revised text to make it more clear.

Most importantly, we did test our method on purely Gaussian and purely tapered power-law distributions, and the obtained values of $\varepsilon$ were correctly predicted as close to zero and close to one, respectively – that is, the method did "recognize" extreme cases when only one part of the two distributions was present in the data (we admit that we should have written that in the first version of our manuscript; we added that information in the revised version). Also, we used the results of our Monte carlo simulations to calculate the $p$-values suggested by the Reviewer (see the revised text). As expected from our previous results, the worst results in terms of the metrics that we used were obtained for test 1450, for which the tapered power law without the Gaussian component seems better than the model with both components.

Also, we would like to mention that the number of adjustable parameters $n_p = 5$ is just higher by 1 in comparison to the popular "double power law" fits proposed by Toyota and other authors (two power law exponents + the location of the regime shift + the minimum floe size for which the distributions are fitted). A small number of adjustable coefficients is often possible only due to the fact that only the tail of the pdf is analyzed, as is the case of the procedure described by Virkar and Clauset (2014; $n_p = 2$). We analyze the whole range of the floe size values.

Finally, in our opinion the statement that the pdfs we are proposing "have little connection to physics" is somewhat exaggerated. Of course, more data, both from the field and from the lab are necessary to confirm or reject the proposed distribution as suitable (and to define a range of conditions under which it is suitable), but both the Gaussian and, especially, the tapered power law are well established in physics and modeling of fracture – as we describe in detail in the discussion. Moreover, as suggested by the other reviewer, we estimated the location (relative to the ice edge) of the maximum bending stress from mechanical properties of the ice, and – as we write in the revised discussion – the mean of the Gaussian part of the pdfs shows a very good agreement

with those values, indicating that the dominant floe size might result from that breaking mechanism.

*14. Page 9 - Discussion. How does one explain the relatively rectangular character of the ice here, relative to the relatively circular character of the ice in real conditions? Is the grinding of floes a significant factor in this?*

Ice floes formed by wave breaking are not circular, but angular in real conditions as well. See, for example, Fig. 1 in Squire (1984), Fig. 1 in Langhorne et al. (1998), or Fig. 1 in Squire and Montiel (2015), which contains snapshots from a time-lapse movie showing two cases of wave-induced breakup, recorded by Dany Dumont and available online at https://vimeo.com/106835989 (see the fragment between 1:15 and 1:35). In general, there are many images of approximately rectangular sea ice floes, but unfortunately most lack any information about their evolution. https://photolibrary.usap.gov/PhotoDetails.aspx?filename=SEAICEBLOWNOUT1961.JPG is a nice example. We added some comments on that to the last part of the discussion section.

And yes, in our opinion grinding (especially under shear deformation) is a mechanism that is responsible for approximately circular floe shapes.

*15. I would like to see a plot of something like mean floe size in time (or as a function of breaking event ), as this is of importance for models of the FSD.*

There are only 3 values from Test Group A, and 2 values from Test Group B, too little to make a plot. They are provided in Table 2.

*16. The discussion of the sum-of-two-distributions idea is well-taken, but a discussion of other processes that act on floes other than those influenced by waves would be good to have, in particular the fact that these processes may or may not be dominant regionally or hemispherically (i.e. for example, if in the Southern Ocean, waves are important but not in the Arctic).*
We do agree that those issues are both very interesting and important, but in our opinion it is beyond the scope of this paper. We intentionally tried to limit the discussion to processes relevant to wave–ice interactions, that is, those that we could observe in our experiments – as the title of the paper clearly says. We do not intend to write a review of all processes that may shape the floe-size distribution under different conditions and in different types of the ice cover. In our opinion, this would obscure the results of this particular study.

**REPLY TO THE COMMENTS OF REVIEWER No. 2**

*1. Regarding the fitting functions, could you explain why you consider these two types of functions (the gamma function and the normal distribution) are appropriate for representing the wave-induced FSD? I could understand from the description in Discussion section that there seems to be two different processes and it would be appropriate to represent the FSD by a summation of two different functions. I can understand one should be the normal distribution if there is a representative length. But how do you explain physically about the gamma function? Since this issue seems to be the key of this paper, it might be better to add some brief explanation earlier, e.g. when the two functions were defined in section 3.2.*

We are a bit confused by this comment in combination with the following ones (No. 3 and 4). The comment above suggests that the Reviewer does not have objections regarding our usage of a sum of two functions to represent the FSD. Also, he/she does not have objections regarding the Gaussian component. Only the second component seems to him/her controversial, even though he/she further argues for a power law as a suitable form of a pdf describing floe sizes. The Gamma distribution is a power law! It is a TRUNCATED power law – and based on the further comments of the Reviewer we infer that it is the truncation that the Reviewer finds so "suspicious". As we write in our discussion, there are many reasons why we should expect to observe truncated power laws, even if the underlying processes produce scale invariance (note that we

do not claim that they don't!).

The Gamma distribution is found in critical phenomena in the presence of finite size effects. It is used e.g. in the analysis of earthquakes, which is arguably one of the best known examples of a process exhibiting scale invariance. The Gamma distribution in this case provides a constraint on the magnitude of the largest earthquakes (there are no infinitely large earthquakes; consequently, the pure power law does not fit observations in the range of the largest events). Similarly, no one seriously questions the fact that the sizes of tectonic plates in the Earth's crust exhibit scale invariance. At the same time, a deviation from a pure power law is observed for the largest plates, resulting from the fact that their total surface area is limited by the available space on the Earth's surface. In exactly the same way, it is not possible to observe a power law tail in fragment sizes resulting from breaking of a finite-size, rectangular plate – as in the case of our experiments. The sizes of the largest ice floes in this case have to deviate from the power law, even if the processes "producing" the floes are scale invariant. Once more: by using the Gamma distribution we do not say they are not!

In the context of sea ice, the same type of a pdf was used e.g. by Weiss (2013) to describe the pdfs of shear stress observed during the SHEBA experiment (see his Fig. 4.4). More importantly for this discussion, Gherardi and Lagomarsino (2015) discuss in detail statistical models of fragmentation to explain their FSD data. All those models predict an exponential cut-off, present in the observational data. We added that information to the revised manuscript.

Finally, note also that the Weibull distribution, used by Lu et al. (2008) and mentioned by the Reviewer in one of the following comments, has the same general form: it is a product of a power-law term and an exponential term. (See their eq. (6): the mean of their distribution is expressed in terms of the Gamma function. Those two pdfs are closely related.)

As for the last part of this comment, i.e., providing an explanation for this particular
choice of functions earlier in the text: we purposefully avoided mixing the part related to the data analysis and that related to interpretation/discussion etc. But, following the suggestion of the Reviewer, we added a sentence (after Eqs. (4) and (5)) saying that a detailed discussion can be found in Section 4, so that the reader knows that he can find an explanation further in the text.

*2. Besides, how do you explain the representative size of the normal distribution? The value of about 0.5 m observed in the tank is clearly different from the wavelength. As the value of 0.5 m is almost common for all the experiments, it might be related to the ice property such as minimum buckling size determined by the stiffness of the material (Mellor, 1986). Please try to find a reasonable explanation. I think this is important to apply the result to the real FSD in sea ice.*

Thank you for pointing this out! We should have tried to estimate the location $x_m$ of the maximum bending moment – it turns out that it provides a nice explanation for our results! When we estimate that location using the formula of Mellor (1986) and the values of $E$ and $h_{\text{ice}}$ from measurements, we obtain $x_m = 0.48$ m for Test Group A and $x_m = 0.62$ m for Test Group B. We added details about that to the discussion section.

*3. Regarding the description "scale-invariance of floe size distribution is assumed a priori" (P2L3), and "In many cases, no convincing arguments for assuming power-law FSDs exist" (P2L7-8), I do not agree. Although I might be wrong, to my understanding, this is not necessarily true. I understand researchers did not assume scale-invariance a priori, but examined it by showing how well a power-law function fits the FSD observed. Personally, I consider there is a good reason for thinking of a power-law as a candidate of PDF function. In nature, it would be natural that size distribution (for any material) tends to become scale invariant without any external forcing that determines the scale. Historically, it has been well known that there is a scale invariant property in the process of sea ice breakup (Weiss, 2001 Engineering Fracture Mechanics Vol.68). In this experiment, a scale given by external forcing would be wavelength. Therefore, it might be possible that the FSD for floes smaller than the wavelength becomes scale*

*invariant through the breakup process and follows a power-law function. Also in your results, there is a possibility that floe size may have a scale invariance property, judging from Fig.2, although you asserted "do not look fractal" in P9L22. (For example, I recommend you plot long-axis against short-axis of the ellipse for individual floes. It might show almost linear relationship.)*

We answered many of these questions earlier in reply to the first comment. Once more: by using the truncated power law we do assume scale-invariance, but we also take into account all kinds of finite-size effects present in our data.

*4. Besides, it should be kept in mind that in reality the FSD formation process is not limited to the wave-induced breakup of sea ice. The herding and other processes that do not have specific scales, which is induced by winds and/or currents, can also contribute to the FSD formation as pointed out by Toyota et al. (2011). Therefore, when the controlling scale is unclear, I do not think it is unreasonable to try to use a power-law function for fitting to test the scale invariance. What do you think? (But as far as this experiment is concerned, I agree that the normal distribution should be used because regular waves with a fixed wavelength can determine the dominant scale.) Thus, I recommend you reconsider about this matter.*

Of course, unquestionably there are many mechanisms shaping the FSD under different conditions. We do not write it is "unreasonable to try to use a power law function" for fitting those FSDs! Again: we do use a power law as well. But, once more, even if scale invariance is observed, there are deviations from power laws at large floe sizes due to all kinds of finite-size effects, some related to the finite size of the domain, other related to the physical nature of the processes involved – as we discuss in detail in our discussion.

*Specific comments:*

*1. (P2L8-9) "no alternative pdfs are considered" This is not necessarily true. For example, Lu et al. (2008) used the Weibull distribution for fitting.*

The Reviewer seems to have overlooked the word "Typically" at the beginning of this sentence. Yes, there are exceptions, but in fact very few.

*2. (Section 2) Please describe whether you used pure ice or sea ice in the experiment. It would provide useful information to consider the ice flexural strength.*

As suggested by both Reviewers, we added the information about salinity to Section 2.1.1.

*3. (Table 1) How did you obtain the wavelength? Was it measured directly or estimated from the theory of deep water approximation?*

The values in Table 1 are calculated from open-water formulae (taking into account the finite water depth). Our analysis of the sensor data from the LS-WICE experiment shows that the wavelengths in the ice were very close to open-water ones, within 0.95–1.05 range. We added that information at the end of Section 2.1.1.

*4. (Table 2) Please add the explanation of Nf and Nf,all in the caption rather than in the text.*

We added this information to the table footnote.

*5. (Section 4) It would be desirable if you include some description about how these results can be applied to the real FSD of sea ice.*

We don't have a separate paragraph on that (in order not to repeat the same things twice), but most issues raised in the discussion are related to "real" sea ice just as they are related to lab experiments. For instance, the problem raised by the Reviewer – about how the dominating size of the floes is related to mechanical properties of the ice – is relevant for both "real" and lab conditions. Similarly, finite-size effects are present in the satellite/airborne data (with some additional sources of those effects, related to limited size of the images etc.) and they can be accounted for by the same methods as in the case of lab data.

Please also note the supplement to this comment:
https://www.the-cryosphere-discuss.net/tc-2017-186/tc-2017-186-AC1-
supplement.pdf

**Supplement:**

[revised manuscript text omitted]